# Influence of Sludge Initial pH on Bioleaching of Excess Sludge to Improve Dewatering Performance

**Shaonan Lin [1,†], Mingyan Shi [1,†], Jiade Wang [1], Huijie Zhu [2,\*] and Guicheng Wen [1]**

[1]   School of Civil Engineering, Guangzhou University, Guangzhou 510006, China; l839146093@163.com (S.L.); mingyanshi@163.com (M.S.); c17362630865@163.com (J.W.); WGC15113675092@163.com (G.W.)
[2]   School of Civil Engineering, Luoyang Institute of Science and Technology, Luoyang 471000, China
\*   Correspondence: huijiezhu@lit.edu.cn
†   These Authors equally contributed to the work.

**Abstract:** pH has an important effect on the physiological activity of eosinophilic microorganisms. Therefore, this study used excess sludge produced by the mixed treatment of leachate and municipal sewage to explore the impact of different sludge initial pH on microbial biochemical reactions associated with the performance of excess sludge dehydration. Shake-flask tests were performed using inoculated microorganisms and fresh excess sludge in 500 mL Erlenmeyer flasks at a ratio of 1:4, with the addition of 2 g/L $S^0$ and 6 g/L $FeS_2$ as energy sources. Erlenmeyer flasks were shaken for 72 h at 180 rpm and 28 °C, in a reciprocating constant homeothermic oscillating water-bath. Results show that the specific resistance to filtration (SRF) of the bioleached excess sludge decreased from $(1.45 \sim 6.68) \times 10^{12}$ m/kg to $(1.21 \sim 14.30) \times 10^{11}$ m/kg and the sedimentation rate increased from 69.00~73.00% to 81.70~85.50%. The SRF decreased from $1.45 \times 10^{12}$ m/kg to $1.21 \times 10^{11}$ m/kg and the sedimentation rate increased from 69.00% to 85.00%, which both reached the highest level when the initial pH of the excess sludge was 5 and the bioleaching duration was 48 h. At this time, the rates of pH reduction and oxidative redox potential (ORP) reached the highest values (69.67% and 515 mV, respectively). Illumina HiSeq PE250 sequencing results show that the dominate microbial community members were *Thiomonas* (relative abundance 4.59~5.44%), which oxidize sulfur and ferrous iron, and *Halothiobacillus* (2.56~3.41%), which oxidizes sulfur. Thus, the acidic environment can promote microbial acidification and oxidation, which can help sludge dewatering. The presence of dominant sulfur oxidation bacteria is the essential reason for the deep dehydration of the bioleached sludge.

**Keywords:** initial pH; excess sludge; bioleaching; dehydration

## 1. Introduction

With the acceleration of industrial production processes and the improvement of living standards, sewage sludge production is increasing rapidly in China. As of 2018, 5222 sewage treatment plants have been built in Chinese cities, with a processing capacity of 228 million m³/day. However, as a by-product, excess sludge production has reached 56.65 million tons (80% water content) and this value is expected to reach 61.77 million tons in 2020, with a compound annual growth rate of about 4% in the next three years [1]. Because sludge has a high water content and large volume, dehydration must be carried out to reduce sludge volume to facilitate transportation and subsequent treatment and disposal. Sludge, which is a hydrophilic colloidal particle with a high bound water content, is very difficult to dehydrate [2]; therefore, to achieve deep dewatering, preconditioning must be carried out. Currently, commonly used methods include freezing and thawing [3]; ultrasound [4]; thermal hydrolysis [5]; and chemical coagulation [6]. Although these methods have certain treatment effects, there are still varying degrees of limitations. For example, freezing and thawing is most suitable for cold climates [7]; and

ultrasound and thermal hydrolysis have high energy requirements and costs [8]. Finally, chemical coagulation requires the addition of a conditioning agent that introduces a large amount of inorganic substances into the sludge and reduces the organic matter content of the sludge dry matter, which is not conducive to subsequent resource utilization [9–11]. Therefore, the development of efficient and economical sludge deep dewatering technologies has become one of the main research directions in this field.

Bioleaching technologies mainly use biochemical reactions of leaching microorganisms to improve the sludge dewatering performance [2,12]. pH reportedly plays an important role in the growth and reproduction of microorganisms [13,14]. Most of the existing research has focused on the treatment of municipal sewage sludge sourced from domestic sewage. However, due to the limited independent disposal capacity of leachate in some landfills in China, it is necessary to mix landfill leachate with municipal sewage [15,16]. Domestic sewage interfused with landfill leachate will cause an increase in toxic and harmful substances within the municipal sewage sludge. The microorganisms subsequently secrete extracellular polymeric substances (EPS) during cell metabolism [4,17–19], which may cause foam during sewage treatment and will affect the normal operation of the system and increase the difficulty of dehydration [20,21]. The dehydration performance in different acid-base environments may vary.

Therefore, based on the previous research into the bioleaching treatment of municipal sludge, this research group tried to apply bioleaching technology to the dehydration of the residual sludge generated by the mixed treatment of leachate and municipal sewage, no relevant literature has been reported. This study used excess sludge produced by the mixed treatment of leachate and municipal sewage to explore the impact of different sludge initial pH (from 3 to 10) on microbial biochemical reactions associated with excess sludge dehydration performance. Following high-throughput Illumina HiSeq PE250 sequencing, microbial community structure and physiological function were analyzed to provide a reference for the comprehensive development of deep dewatering technologies for excess sludge.

## 2. Materials and Methods

### 2.1. Original Sludge Properties

The original sludge used in this study was obtained from the secondary clarifier of an anaerobic-anoxic-oxic ($A^2$/O) process in the Datangsha Sewage Treatment Plant (Guangzhou Province, China) and the process influx was mixed with 0.13% landfill leachate. The landfill leachate comes from Guangzhou Xingfeng Landfill (Guangzhou, China), the main components are shown in Table 1.

**Table 1.** Characteristics of the landfill leachate.

| Parameters | Value |
|---|---|
| pH | $6.00 \pm 1.00$ |
| Chemical Oxygen Demand(COD)/(mg L$^{-1}$) | $15{,}800 \pm 7200$ |
| Ammonia-Nitrogen(NH$_3$-N)/(mg L$^{-1}$) | $4050 \pm 1650$ |
| Total Nitrogen (TN)/(mg L$^{-1}$) | $4350 \pm 1750$ |

Three samples were taken in parallel under the same conditions. After collection, the sludge samples were refrigerated at 4 °C. The physiochemical properties of the collected original sludge are listed in Table 2.

**Table 2.** Physicochemical characteristics of the tested sewage sludge.

| Parameters | Value |
|---|---|
| pH | 6.83 ± 0.21 |
| Solid content % | 1.32 ± 0.32 |
| Sludge sedimentation rate % | 61 ± 4 |
| Specific Resistance to Filtration (SRF)/(mg L$^{-1}$) | $(4.44 \pm 0.39) \times 10^{12}$ |

## 2.2. Bioleaching Experiments

The pH of the original sludge adjusted to 3.0, 5.0, 6.9, 8.0 and 10.0 by using 25% *v/v* hydrochloric acid or 10% *v/v* sodium hydroxide, and then it was used as the tested sludge for shake flask test. The initial pH mentioned in the paper refers to the initial pH of the tested sludge. Tests were performed for 72 h. The inoculum and fresh sludge were added to a 500 mL Erlenmeyer bottle at a ratio of 1:4, followed by the addition of $FeS_2$ (6 g/L) and $S^0$ (2 g/L). Erlenmeyer flasks were shaken at 180 rpm and 28 °C in a reciprocating constant homeothermic oscillating water-bath (SHA-C). The oxygen concentration in sludge is 1 mg/L.

Under the optimal pH obtained from the experiment, the blank experiment was carried out, that is, after adjusting the pH value of the tested sludge, the shake flask test was conducted directly without adding inoculum.

The sludge sample of $Fe^{2+}$, $Fe^{3+}$, $SO_4^{2-}$ and TOC were centrifuged at high speed, the supernatant was taken, and filtered by a 0.45 μm filter membrane. The main components of EPS are polysaccharides and proteins, whose TOC accounts for 70~80% of the whole EPS. In this paper, the unit of EPS content is mg/g (denoted by VSS), so it is expressed as TOC/VSS. The tested parameters and analytical methods are shown in Table 3. The correlation between indexes was analyzed by SPSS software.

**Table 3.** Test parameters and analytical methods.

| Parameter | Method |
|---|---|
| pH, ORP | MultiHQ40d portable pH/DO tester |
| $Fe^{2+}$, $Fe^{3+}$ | Phenanthroline spectrophotometry [22] |
| $SO_4^{2-}$ | Barium chromate spectrophotometry [23] |
| Specific Resistance to Filtration (SRF) | Buchner funnel test [24] |
| Sludge particle size | Mastersizer 2000 type laser particle size analyzer |
| Moisture content | Drying at 105 °C for 24 h and then weighing [25] |
| Volatile Suspended Solids (VSS) | Burning in a muffle furnace at 600 °C for 0.5 h and then weighing [26] |
| Total Organic Carbon (TOC) | TOC-VCPH type total organic carbon analyzer |
| EPS (replace by TOC and VSS ratio) | Heat extraction [27] |
| Sludge sedimentation rate | Gravity separation method |

## 2.3. Preparation of Inoculum

The inoculum was prepared in 250 mL Erlenmeyer flasks containing 150 mL of fresh sludge spiked with 10 g/L $S^0$ and 10 g/L $FeS_2$ as the energy source. Erlenmeyer flasks were shaken at 180 rpm and 28 °C in a reciprocating constant homeothermic oscillating water-bath (SHA-C). Changes in sludge pH were monitored every 12 h. When the pH was below 2.5, the first culture was considered period over [28]; then 20 mL of the bio-acidified sludge was transferred to a new flask and combined with 130 mL of fresh sludge. The above process was repeated twice. The obtained acidified sludge then met the inoculum requirements.

## 2.4. Microbial Community Analysis

### 2.4.1. DNA Extraction and PCR Amplification

Three samples were taken in parallel under the same conditions. DNA from one replicate of each sludge sample was extracted using the PowerSoil™ DNA Isolation Kit (MO BIO). The extracted

DNA was stored at −20 °C (if it could not be immediately amplified by PCR, it was stored at −80 °C). The quantity and quality of the extracted DNA were assessed by NanoDrop spectrophotometry (Thermo Scientific, MA, USA) and 1.5% agarose gel electrophoresis.

DNA from the extracted sludge sample was used as a template. Three PCR amplifications were performed in parallel. The V4-V5 hypervariable region of the 16S rRNA gene was amplified using the 16S rRNA universal primers 515F (5′-GTGCCAGCMGCCGCGGTAA-3′) and 907R (5′-CCGTCAATTCMTTTRAGTTT-3′). The thermocycling steps were as follows: 94 °C for 5 min; followed by 31 cycles at 94 °C for 30 s, 52 °C for 30 s, 72 °C for 45 s; and a final extension step at 72 °C for 10 min. Amplicons were held at 4 °C or stored at −80 °C if they could not proceed to the next step immediately; Triplicate analyses were performed for each sample. Agarose gel electrophoresis was used to detect the PCR products to determine whether DNA was successfully extracted from sludge samples.

### 2.4.2. Illumina HiSeq Sequencing of PCR Products

Using gelatin to recover the PCR product, the amplicons from the three sewage sludge samples were mixed at equal concentrations and terminal sequencing was performed on the Illumina HiSeq PE250 platform. Quantity and quality evaluation of the paired fastq sequences were performed using fastqc software and Perl scripts were used for quality control. A total of 159,533 sequences were obtained, with an average length of 260 bp. Operational taxonomic units (OTUs) were assigned using USEARCH software at a 97% sequence similarity cutoff. The phylogenetic affiliation of each 16S rRNA gene sequence was classified to different taxonomic levels using the RDP classifier, based on a comparison with the 16S rRNA database at a 97% confidence threshold.

### 2.4.3. Data Analysis

a.  Raw data filtering: raw data was mass filtered using Trimmomatic software to obtain quality data after quality control.
b.  Clean data PE reads: raw contigs were obtained by splicing each pair of PE reads using software, including Mothur and Flash.
c.  Raw contig sequence quality filtering: clean contigs were obtained using Mothur for quality control and filtering of stitched sequences.
d.  Clean contig sample allocation: according to barcode and primer information, Qiime software was used to assign the spliced sequences to their corresponding samples.

### 2.4.4. Diversity Calculation

According to the OTU table and phylogenetic tree, the alpha diversity of each sample was calculated via the Chao1 estimator, and Shannon and Simpson indices as per Equations (1)–(3), respectively.

$$H_{shannon} = -\sum_{i=1}^{S_{obs}} \frac{n_i}{N} \ln \frac{n_i}{N} \tag{1}$$

$$D_{simpson} = 1 - \frac{\sum_{i=1}^{S_{obs}} n_i(n_i - 1)}{N(N-1)} \tag{2}$$

where Sobs is the measured number of OTUs; $n_i$ is the number of sequences contained in the *i*-th OTU; and *N* is the number of all sequences.

$$S_{Chao1} = S_{obs} + \frac{n_1(n_1 - 1)}{2(n_2 + 1)} \tag{3}$$

where, $S_{Chaol}$ is the estimated OTU number; $n_1$ is the number of OTUs containing only one sequence; and $n_2$ is the number of OTUs containing only two sequences.

## 3. Results and Discussion

### 3.1. Effect of Different Sludge Initial pH Values on Acidification and Oxidation during Sludge Bioleaching

#### 3.1.1. Effect of Different Sludge Initial pH Values on Acidification

The changes in pH during the bioleaching experiments under different initial pH conditions are shown in Figure 1.

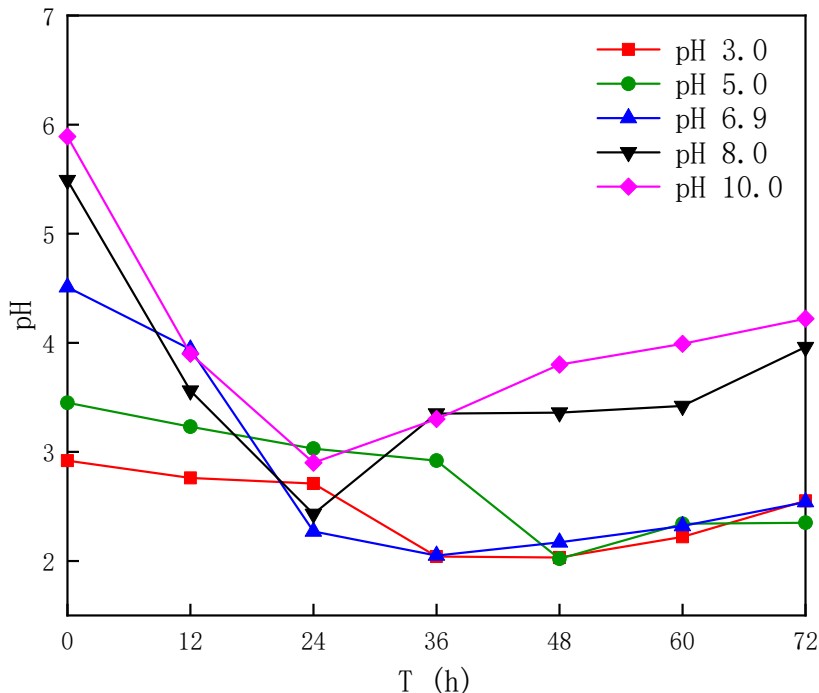

**Figure 1.** Influence of initial pH on the dynamics of the pH value during the process of bioleaching.

When the initial pH was 3.0, 5.0, 6.9, 8.0 and 10.0, the maximum rates of pH reduction were 69.52%, 69.67%, 70.07%, 63.18% and 56.06%, respectively (Figure 1). Thus, relatively more acid is produced under acidic conditions. The possible reason for this is that bioleaching microorganisms are mainly made up of weak eosinophilic microorganisms and strong eosinophilic microorganisms, which have an optimum pH of 4.0~5.0 and 1.4~2.2, respectively [28,29]. The pH was controlled in the range of 3.0 to 10.0, because the extreme acid-base environment outside this range will corrode equipment and increase the treatment cost. At the beginning of bioleaching, weakly eosinophilic microorganisms grow rapidly at the neutral pH, which promotes the decrease in pH, when the pH drops to about 4.0, the growth stops. Under acidic pH conditions, strong eosinophils begin to proliferate, which subsequently decreases the pH further [29], until it eventually reaches 2.02~2.90.

#### 3.1.2. Effect of Different Sludge Initial pH Values on Oxidation

Figure 2 illustrates the $Fe^{2+}$, $Fe^{3+}$, and $SO_4^{2-}$ concentration dynamics during bioleaching.

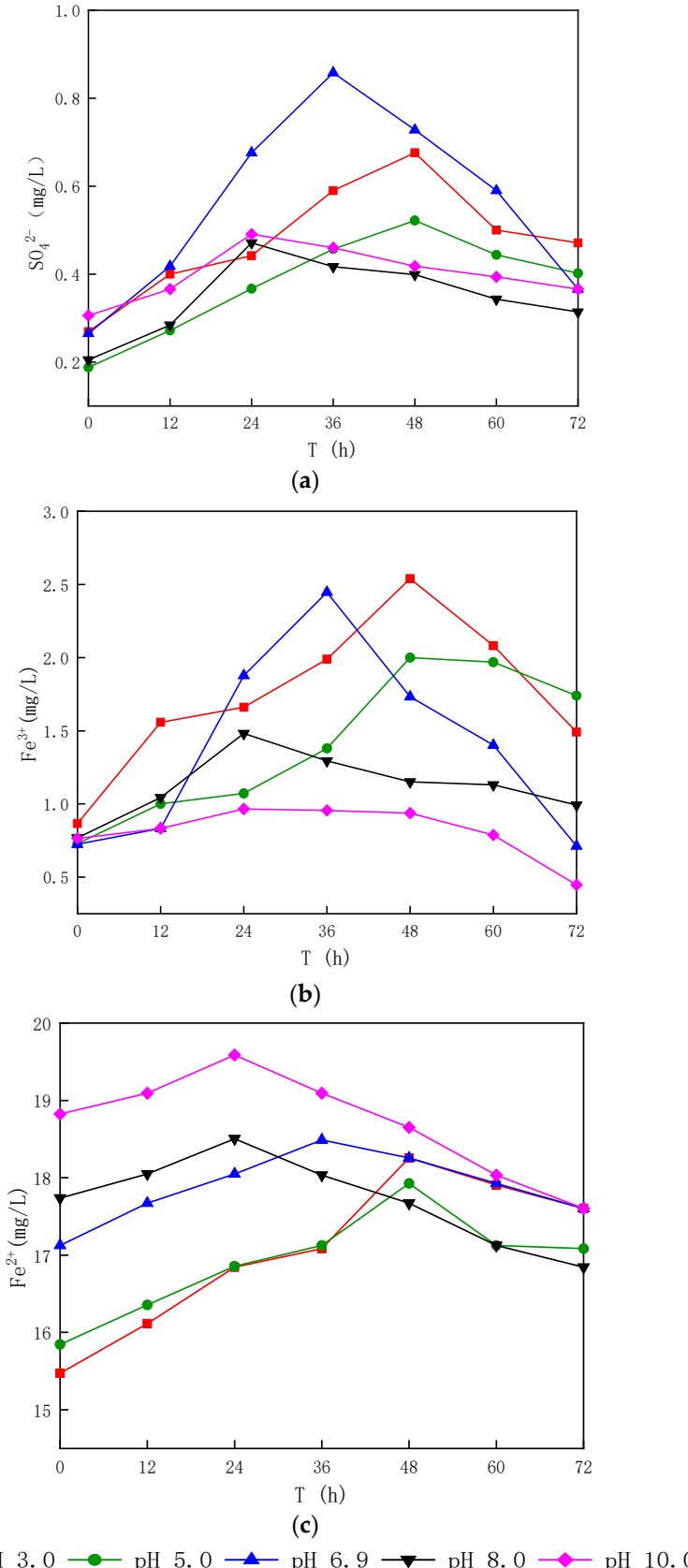

**Figure 2.** Influence of initial pH on (**a**) $SO_4^{2-}$, (**b**) $Fe^{3+}$ and (**c**) $Fe^{2+}$ during the bioleaching process.

During the early stages of the experiment, the highest bioleaching efficiency was achieved and $Fe^{2+}$ concentrations increased to varying degrees (Figure 2). When the initial pH was 3.0, 5.0, 6.9, 8.0 and 10.0, the highest growth rates were 15.24%, 11.62%, 7.38%, 4.15% and 3.90%, respectively. The increase under acidic conditions was about 3.48~11.09% higher than under alkaline conditions. The concentration of $Fe^{3+}$ also showed the same trend, with a maximum increase of 20.91%~70.37%. According to Equation (4), an increased concentration of $H^+$ accelerates the oxidation of $Fe^{2+}$, resulting in a decreased $Fe^{2+}$ concentration and an increased $Fe^{3+}$ concentration. However, when the concentration of $Fe^{2+}$ in the sludge solution decreases, the separation of $Fe^{2+}$ from $FeS_2$ further increases (Equation (5)). Therefore, the increase of $Fe^{2+}$ and $Fe^{3+}$ concentrations is higher under acidic conditions. In the later stage of bioleaching, the concentrations of $Fe^{2+}$ and $Fe^{3+}$ decreased. The possible reason for this is that the energy substance is largely consumed, the microbial activity deteriorates, and the reaction of $Fe^{3+}$ with $SO_4^{2-}$ generates schwertmannite, which consumes a part of $Fe^{3+}$ [30].

$$4Fe^{2+} + O_2 + 4H^+ \rightarrow 4Fe^{3+} + 2H_2O \tag{4}$$

$$FeS_2 + 7/2O_2 + H_2O \rightarrow Fe^{2+} + 2SO_4^{2-} + 2H^+ \tag{5}$$

In the early stage of bioleaching, the $SO_4^{2-}$ concentration also showed an upward trend, which was caused by the oxidation of sulfur to sulfuric acid [31]. When the initial pH was 3.0, 5.0, 6.9, 8.0 and 10.0, the proportions of reductive sulfur oxidized were 60.21%, 63.98%, 69.11%, 56.48% and 37.69%, respectively. It can be seen from the above results that the alkaline condition was relatively low. It is speculated that this is related to the ability of sulfur-oxidizing bacteria to grow in acidic and neutral conditions [32,33]. During the final stages of bioleaching, the $SO_4^{2-}$ concentration decreased. The synchronous decrease of $SO_4^{2-}$ and $Fe^{3+}$ concentrations was related to the decrease of microbial activity in the late leaching stage.

The ORP changes during treatment processes are shown in Figure 3.

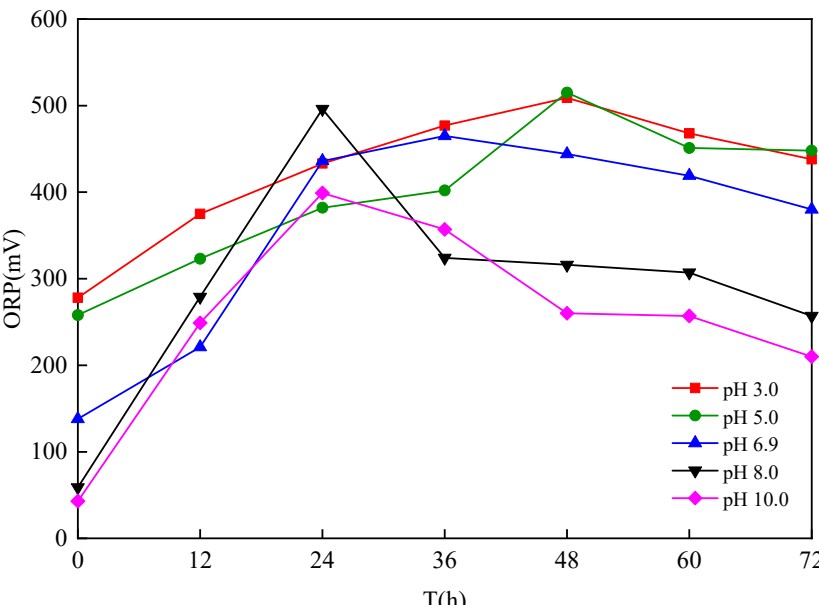

**Figure 3.** Influence of initial pH on the oxidative redox potential (ORP) during the bioleaching process.

During the early stage of leaching, the ORP continuously increased, with maximum values of 399 mV~515 mV, it relatively high at an initial pH of 5 (Figure 3). $SO_4^{2-}$ and $Fe^{3+}$ concentrations increased synchronously with the ORP in the same period (correlation coefficients $R^2$ are 0.9072 and 0.9013.); ORP represents the oxidation ability of microorganisms, which suggests that the change of

ORP is caused by the oxidation of $S^0$ and $Fe^{2+}$ [34]. Therefore, in the later stage of leaching, when the concentration of $SO_4{}^{2-}$ and $Fe^{3+}$ drops, ORP also drops.

### 3.2. Change of EPS Content

According to the combination degree between organic matter and cell phase, EPS is categorized as soluble (S-EPS), loosely bound (LB-EPS) or tightly bound (TB-EPS) [35].

The change of EPS content in different treatment processes is shown in Figure 4.

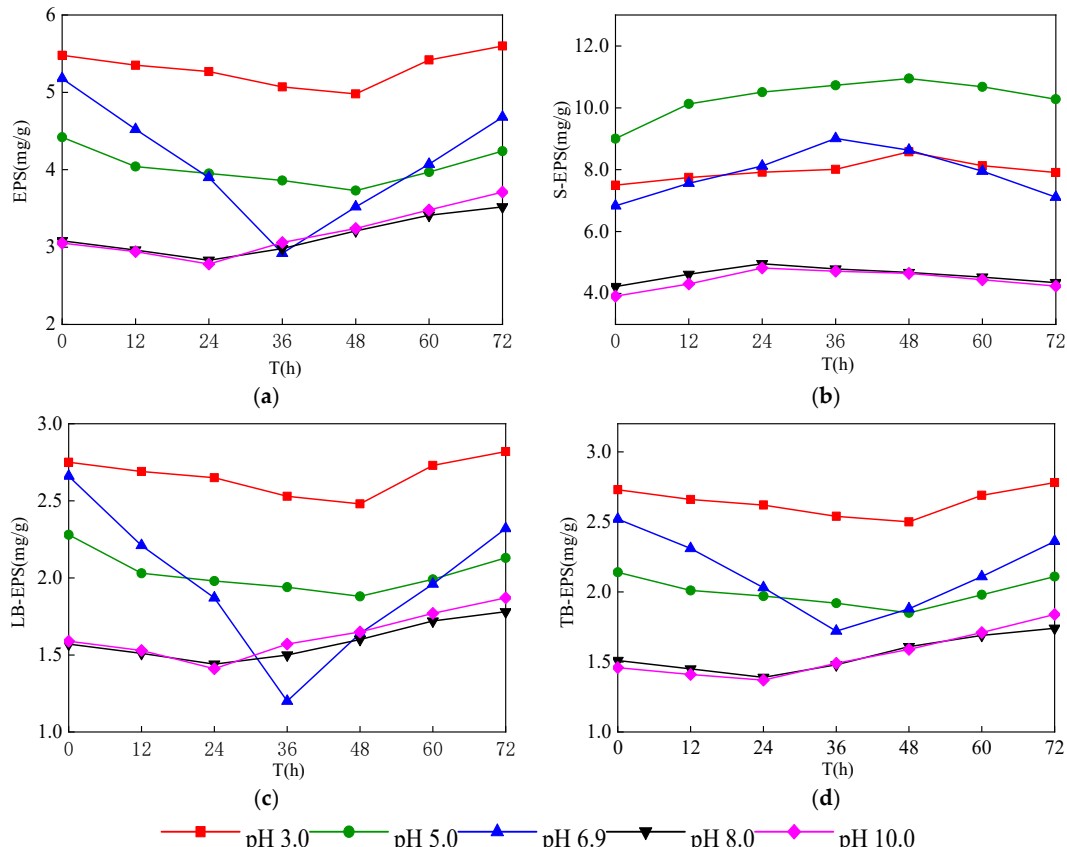

**Figure 4.** Changes of EPS contents during the different treatment processes: (**a**) changes of extracellular polymeric substances (EPS) contents; (**b**) changes of soluble (S)-EPS contents; (**c**) changes of loosely bound (LB)-EPS contents; (**d**) changes of tightly bound (TB)-EPS contents during the different treatment processes.

During the early stage of bioleaching, TB-EPS and LB-EPS continuously decreased to minimums of 9.74~53.52% and 10.56~60.13%, respectively (Figure 4). When comparing with the data from Figure 1, it was found that the pH was synchronized with the changes of LB-EPS and TB-EPS, which suggests that the decreased LB-EPS and TB-EPS content was caused by the microbial acidification reaction. There are two reasons to explain this. On the one hand, the structure of the insoluble proteins in LB-EPS and TB-EPS is destroyed, and they are converted into a soluble state by hydrolysis reactions. On the other hand, a large amount of polysaccharides attached to LB-EPS and TB-EPS are hydrolyzed into glycosides and transferred to the slime layer [36,37], which was evidenced by the simultaneous increase in the S-EPS content.

At this stage, the total amount of EPS also declined, with the largest decrease of 10.16%~50.42%, indicating that the EPS mainly consisted of LB-EPS and TB-EPS. Additionally, the number of heterotrophic microorganisms at this stage decreased sharply (Figure 5), indicating a predominance of autotrophic microorganisms, represented by *Thiobacillus* [38]. Autotrophic microorganisms secrete less EPS [39], which may be one of the reasons for the decrease in total EPS.

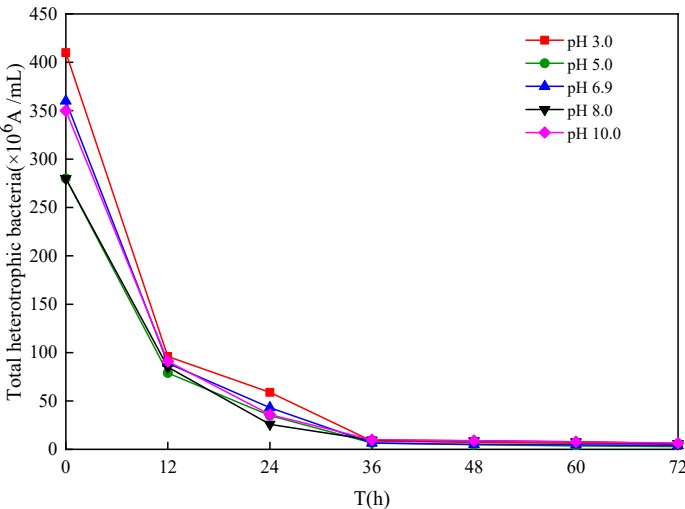

**Figure 5.** The total number of heterotrophic bacteria during different treatment processes.

During the later stage of bioleaching, the total amount of LB-EPS, TB-EPS and EPS increased. This is because microorganisms secrete large amounts of EPS to protect their cells from injury when they are exposed to an acidic environment for too long [35].

### 3.3. Effect of Different Sludge Initial pH Values on Sludge Dewatering Performance

Specific resistance to filtration (SRF) is an indicator of sludge dewatering performance [40]. The degree of sludge dewatering is usually determined according to its range. When SRF > $4.0 \times 10^{12}$ m/kg, it is not easy to dehydrate; when SRF < $1.0 \times 10^{12}$ m/kg, it is easy to dehydrate; when $1.0 \times 10^{12}$ < SRF < $4.0 \times 10^{12}$ m/kg, the dehydration performance is somewhere between the two [41]. The change in SRF during bioleaching is shown in Figure 6.

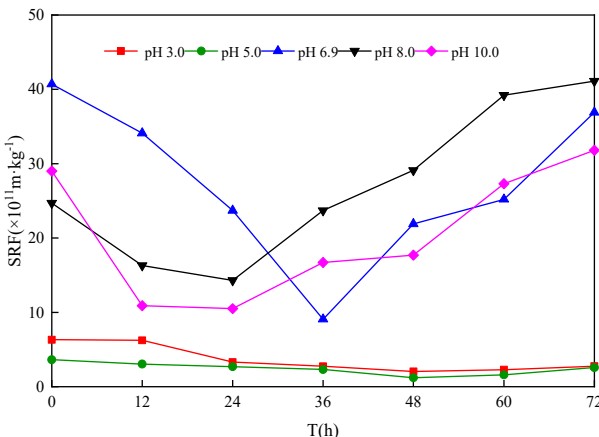

**Figure 6.** Effect of initial pH on sludge specific resistance to filtration during treatment processes.

After the start of bioleaching, the SRF of the sludge continuously decreased, with lowest values from $(0.12\sim1.43) \times 10^{12}$ m/kg (Figure 6). Compared with the original sludge, the SRF decreased by 70.60%~97.03%, thus, the dewaterability of the sludge improved. The results of blank experiment showed that the pH, ORP, SRF and other parameters of sludge hardly changed after adjusting the pH of sludge, and the sludge dewatering performance was not significantly improved. The reasons for this are as follows: (1) TB-EPS and LB-EPS can stabilize the floc structure of sludge and increase its bound water content by connecting microbial cells and other substances. The TB-EPS and LB-EPS content during this stage was continuously decreasing (Figure 4), combined with water release and free water increase, thereby greatly improving sludge dewaterability. (2) The surface of the sludge particles is

surrounded by Zoogloea, forming a hydration shell, which prevents the particles from coagulating and affects sedimentation and dehydration [42]. However, schwertmannite formed during the reaction weakens the hydration by electrical neutralization and adsorption bridging [30]. The Zoogloea also weakens or disappears; the sludge particles destabilize, aggregate and sediment, which improves the dewatering effect [43]. (3) The particle size distribution of the sludge particles (Figure 7) from the optimum bioleaching time at different initial pH values (i.e., when the highest bioleaching efficiency was achieved). The proportion of sludge particles with particle size ≤ 30 μm was 34.17%~42.84%, and the medium-speed quantitative filter paper has a pore size of 30–50 μm. Therefore, the filtration and dehydration performance of the sludge was enhanced [44]. When the initial pH value was 5.0 and the bioleaching time was 48 h, the SRF decrease rate peaked (91.66%), the sludge was easy to dehydrate (SRF = $0.12 \times 10^{12}$ m/kg), and the dehydration performance was the strongest. Correspondingly, the highest proportion of sludge particles, with a particle size ≤ 30 μm (42.84%), and the lowest content of TB-EPS and LB-EPS also support the above inference.

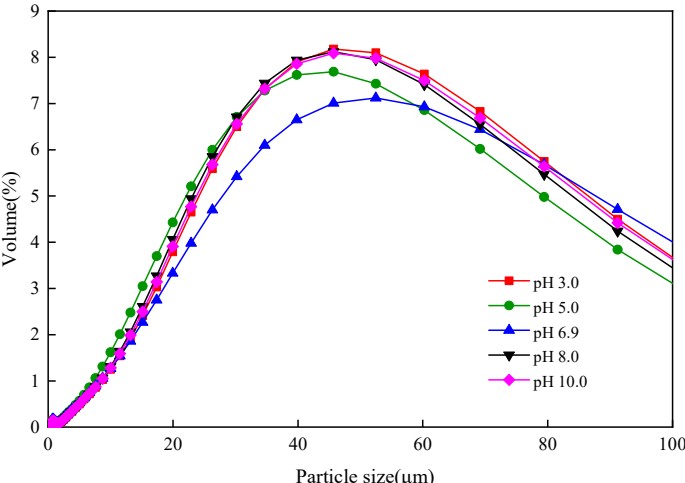

**Figure 7.** The integral distribution curves of sludge particle size during the bioleaching process under different initial pH.

### 3.4. Influence of Different Sludge Initial pH Values on Settling Sludge Performance

Sludge sedimentation performance reflects the gravity concentration performance, which can also be used for indirect evaluation of the sludge dewatering performance [45]. The variation of the settling sludge rate during treatment is shown in Figure 8.

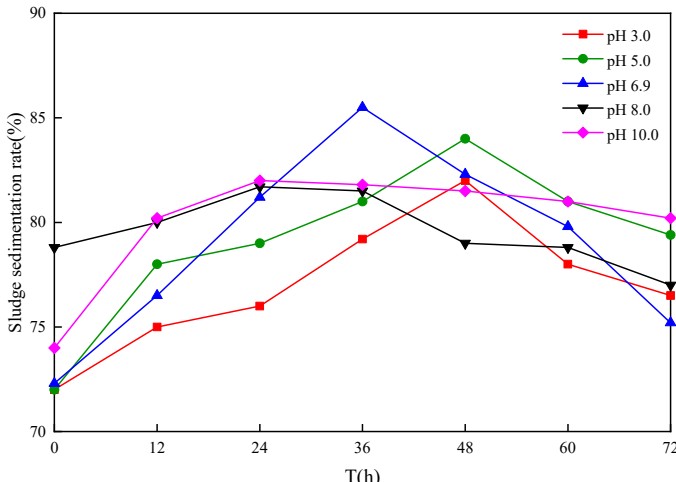

**Figure 8.** Changes of sludge sedimentation rate during different treatment processes.

During the initial stages of bioleaching, the sludge sedimentation rate increased from the initial 69.00~73.00% to the highest value of 81.70~85.50% (Figure 8), at an increase rate of 11.92~22.84%. The increase in sludge sedimentation under acidic conditions was 6.92~10.51% higher than under alkaline condition, which is consistent with the results of demonstrated in Figures 1 and 2. This could be explained in two ways. On one hand, the production of $H^+$ ions by the microbial acidification reaction neutralized the negative charge of the sludge [45], which caused the sludge colloidal particles to destabilize and further collide with each other, forming a tighter floc. On the other hand, $Fe^{3+}$ formed by $Fe^{2+}$ oxidation flocculates and enhances the settling sludge performance [46]. During the later stage of leaching, the sludge sedimentation rate decreased to varying degrees, with the decline rate from 1.80~10.30%, The decreased sludge sedimentation rate occurred due to (1) reduced acid production leading to re-stabilization of the colloidal particle (Figure 4); and (2) at this stage, the concentration of $Fe^{3+}$ decreased, decreasing flocculation (Figure 2). Comparing the SRF of the sludge shown in Figure 6, it was found that the SRF and sedimentation rate of the sludge were synchronized (correlation coefficient $R^2$ is 0.8479), which suggests that bioleaching can simultaneously improve sludge dewatering and sedimentation performance.

### 3.5. Microbial Community Structure

Microbial Community Structure of the Inoculum

Illumina HiSeq PE250 sequencing was performed on the inoculum, and the results are shown in Figures 9 and 10.

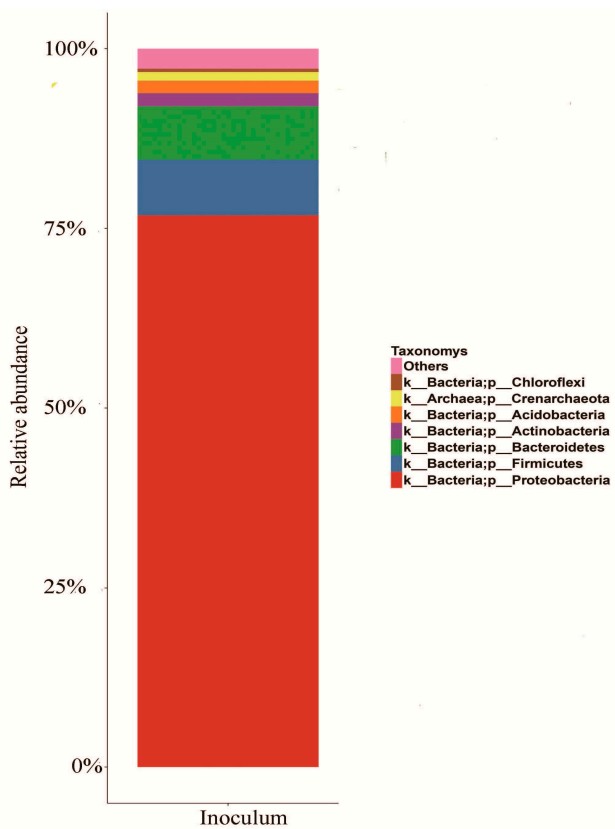

**Figure 9.** Microbial community structure of inoculum at the phylum level.

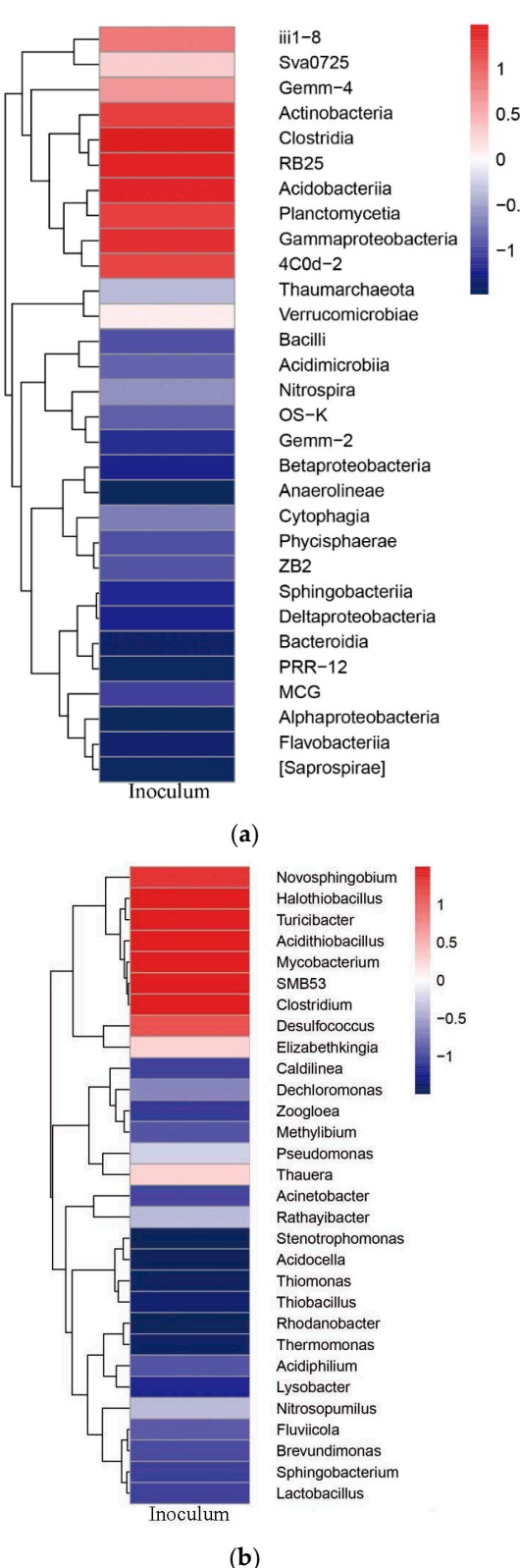

**Figure 10.** Microbial community structure of the inoculum at the (**a**) class and (**b**) genus levels.

At the phylum level, the dominant phylum of the inoculum was *Proteobacteria* (relative abundance 76.84%), followed by *Firmicutes*, *Bacteroidetes*, *Actinobacteria*, and *Acidobacteria*, with the relative abundances from 1.73~7.73% (Figure 9).

The dominant taxa in the inoculum at the class level were *Gammaproteobacteria* (61.79%; of the phylum *Proteobacteria*), which can oxidize sulfide and ferrous [47], followed by *Betaproteobacteria*, *Deltaproteobacteria* and *Alphaproteobacteria*, with relative abundances from 3.10~8.59% (Figure 10).

The dominant microorganisms at the genus level in the inoculum were *Acidithiobacillus* (10.42%), *Thiomonas* (6.92%) and *Halothiobacillus* (2.74%), all of the class *Gammaproteobacteria* (Figure 10). *Acidithiobacillus* and *Thiomonas* can oxidize sulfide and ferrous [48,49], while *Halothiobacillus* can oxidize sulfide [50].

### 3.6. Relationship between Microbial Community Structure Succession and Sludge Initial pH

It can be seen from the above results that acidic conditions are more favorable for sludge sedimentation and dehydration. To that end, the composition and succession of the microbial flora at optimum bioleaching times (48, 48 and 36 h) at sludge initial pH values of 3.0, 5.0 and 6.9 were analyzed using Illumina HiSeq PE250 sequencing technology.

### 3.6.1. Changes of the Microbial Community

When the initial pH was 3.0, 5.0 and 6.9, the microbial community at the phylum level was mainly composed of *Proteobacteria* and *Bacteroidetes*, followed by *Firmicutes*, *Actinobacteria* and *Acidobacteria* (Figure 11). The relative abundance of *Proteobacteria*, which can oxidize $Fe^{2+}$ and reduce sulfur [51], was 73.85~76.97% and the relative abundance of *Bacteroidetes*, which can reduce the pH of sludge by generating sulfuric acid from redox sulfur [52], was 11.60~15.88%. Thus, at the phylum level, the initial pH conditions had no significant effect on the dominant microorganisms.

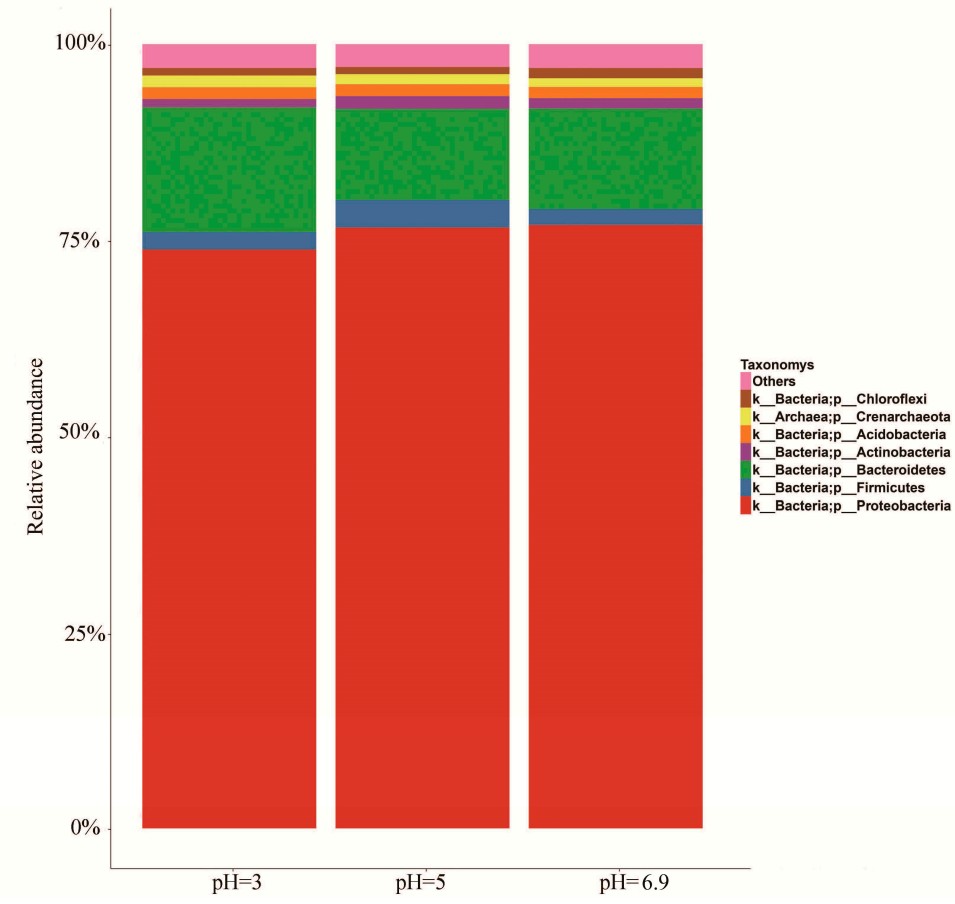

**Figure 11.** Relative abundance of microbial communities at the phylum level.

The relative abundance of the 30 most abundant taxa at the class level are shown in Figure 12. The microbial community of the sludge was mainly composed of *Gammaproteobacteria* and *Betaproteobacteria*, followed by *Alphaproteobacteria*, all of which belong to the *Proteobacteria* phylum. The relative abundance of *Gammaproteobacteria*, which can oxidize sulfide and ferrous iron [47], was 50.99~55.74% and was highest at an initial pH of 5.0. The relative abundance of *Betaproteobacteria*, which has the function of sulfur oxide [53], was 13.27~18.69%. *Alphaproteobacteria* can use iron and sulfur to complete metabolism [53].

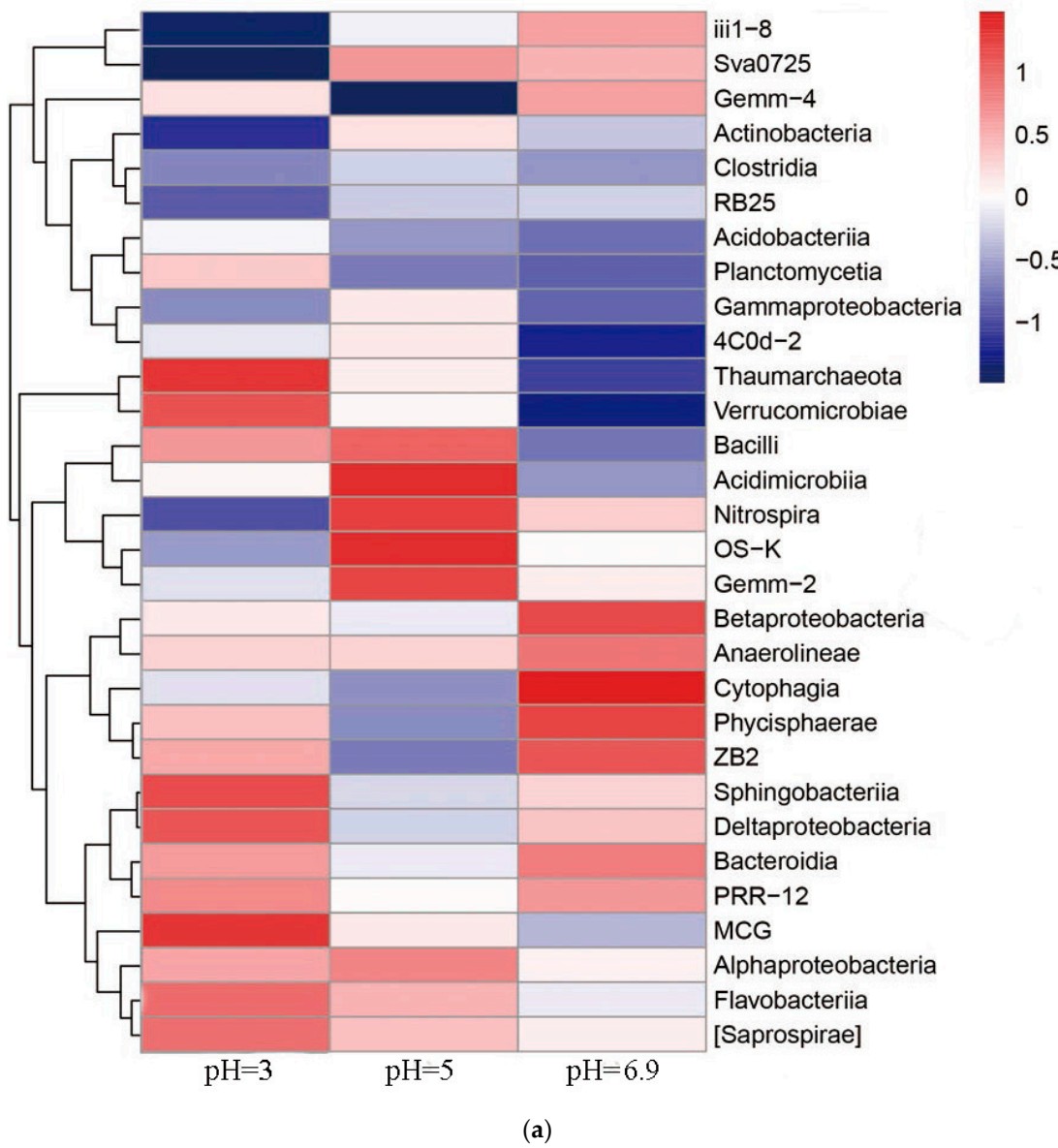

(a)

**Figure 12.** *Cont.*

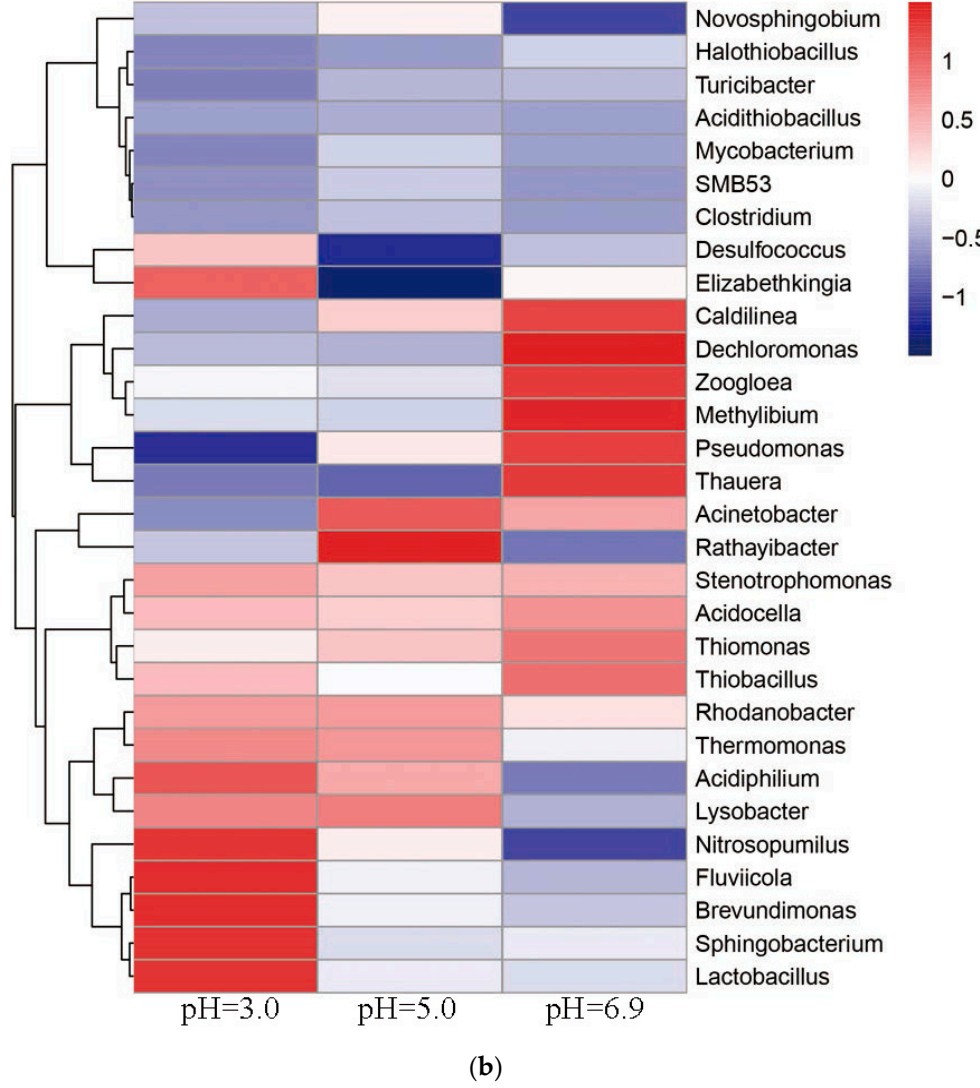

**Figure 12.** Relative abundance of microbial communities at the (**a**) class and (**b**) genus level.

The relative abundance of the 30 most abundant taxa at the genus level are shown in Figure 12. The microbial community at the genus level under acidic conditions was mainly composed of *Thiomonas* of the class of Betaproteobacteria, and *Halothiobacillus* and *Acidithiobacillus* of the class of *Gammaproteobacteria*. Among these, *Thiomanas*, which has the functions of oxide ferrous and sulfur [54], was the dominant flora with relative abundances of 4.59~5.44%. The relative abundances of *Halothiobacillus* and *Acidithiobacillus* were 2.56~3.41% and 0.38~0.80%, respectively. *Halothiobacillus* is capable of oxidizing sulfur [50], while *Acidithiobacillus* has the ability to oxidize sulfur and $Fe^{2+}$ [55].

According to the above analysis, the bacteria in the sludge that have redox sulfur and $Fe^{2+}$ functions are dominant, and are mainly composed of the autotrophic bacteria *Thiomonas*, *Halothiobacillus* and *Acidithiobacillus*. Autotrophic bacteria are dominant. From this, it can be initially inferred that the improved sludge dehydration performance was mainly due to the action of the biological leaching microorganisms. Meanwhile, according to the blank test (Figure 13), the pH decreased from 6.36 to 5.21, the rate of decline was only 18.08%, and the ORP rose from 2 mV to 172 mV. Compared with the bioleaching group, the blank group that, which had only a small amount of sulfur oxidizing microorganisms in the original sludge showed extremely weak acidification and oxidation, thus verifying the above inference.

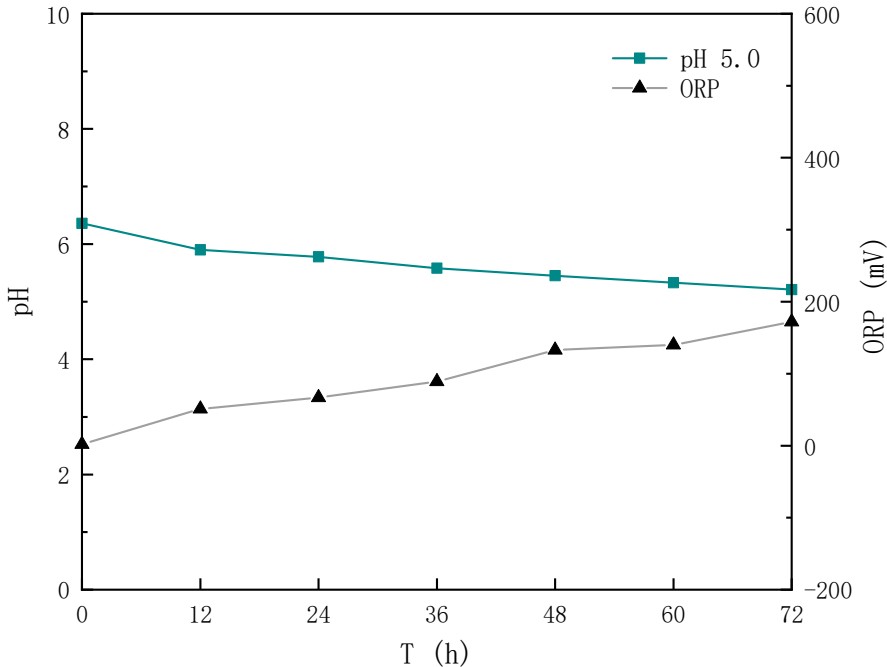

**Figure 13.** Change in pH and ORP in the blank group.

### 3.6.2. Microbial Similarity Analysis

Sequence clustering analysis was carried out for similarity using Venn analysis, and the results are shown in Figure 14.

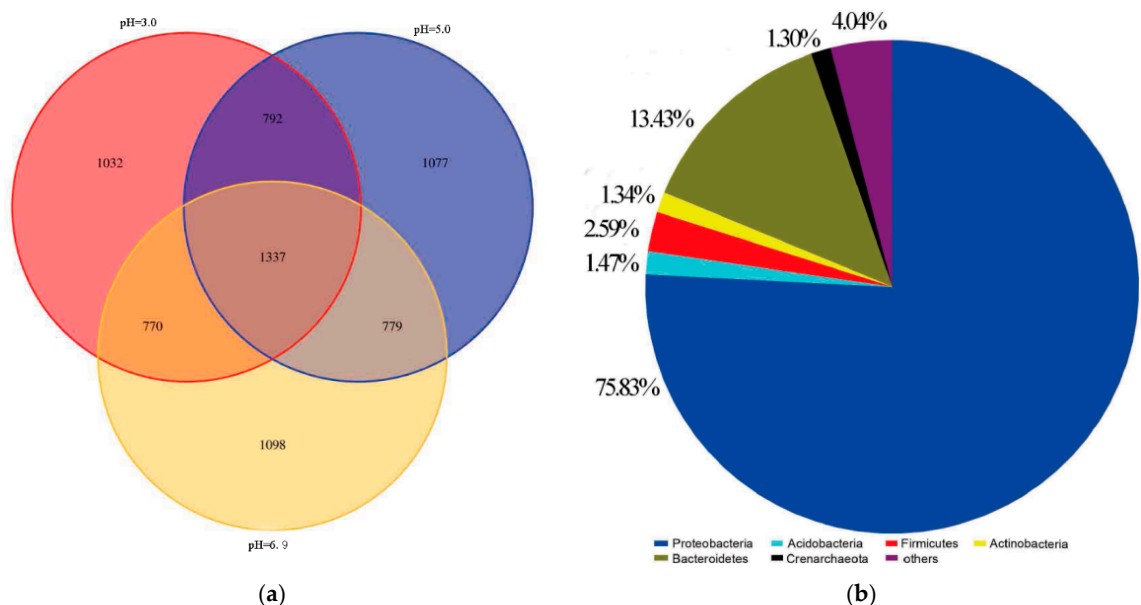

(**a**)      (**b**)

**Figure 14.** (**a**) Venn diagram and (**b**) coincide partial operational taxonomic units (OTU) distribution.

The total number of OTUs for different treatments was 11900, but only 1337 OTUs are common. Among them, 75.83% of the microorganisms belonged to *Proteobacteria*, while 13.43% belonged to *Bacteroidete*. Additionally, the microbial community similarity was higher at the initial pH of 3.0 and 5.0.

### 3.6.3. Analysis of Microbial Community Diversity

The microbial richness and diversity indices of the inoculum and different treated sludge samples are shown in Table 4.

**Table 4.** Richness and diversity estimators of the microbial community under different conditions.

| Simple | Chao1 | Shannon | Simpson |
|---|---|---|---|
| Inoculum | 6880.97 | 8.40 | 0.95 |
| pH = 3.0 | 7822.00 | 8.56 | 0.95 |
| pH = 5.0 | 7972.65 | 8.76 | 0.94 |
| pH = 6.9 | 7585.80 | 8.59 | 0.95 |

The Chao1 estimators of the treated sludge samples is higher than that of inoculum (Table 4), indicating that the bioleaching treatment can increase the relative species abundance. Among the treatments, when the initial pH was 5.0, the Chao1 estimator and Shannon index were the highest, indicating that the microbial community richness and diversity were the highest under this initial pH condition. The Simpson index of the inoculum was close to the different treatments, indicating that species uniformity is independent of the process.

According to the rank-abundance curve (Figure 15), when the initial pH was 5.0, the curve had the longest length on the horizontal axis, which proves that the microbial richness was the highest under this condition. The rank-abundance curves of different treatments were similar in smoothness, which also shows that the species uniformity was similar under the different treatments.

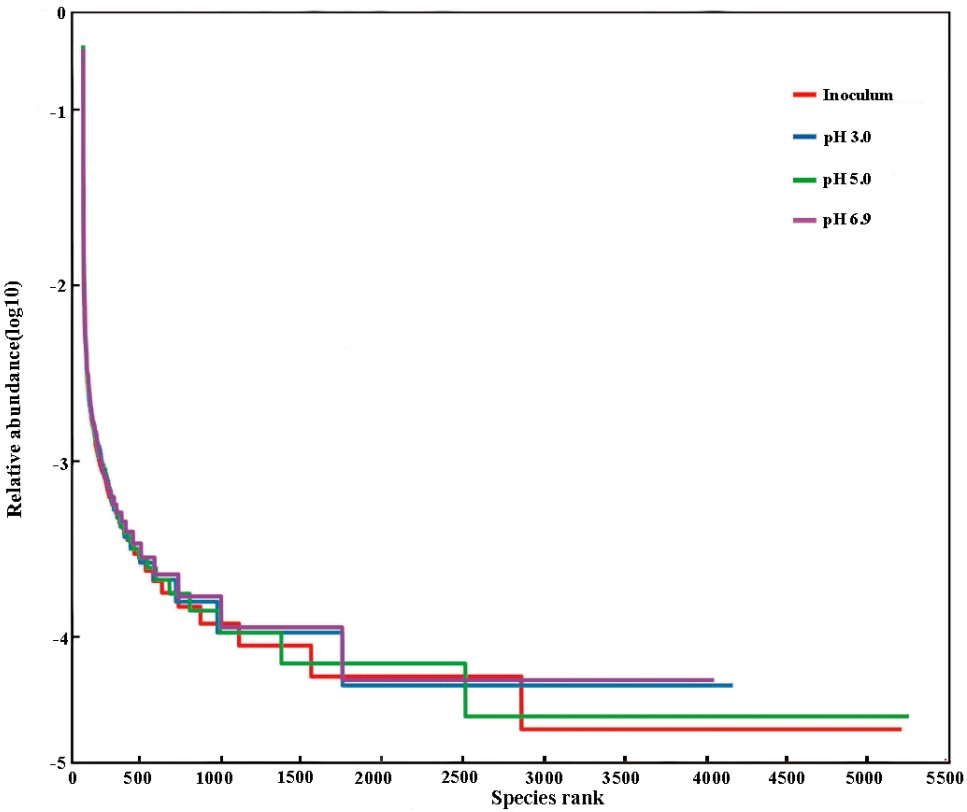

**Figure 15.** Rank–abundance curves of different sludge samples.

## 4. Conclusions

This study, focused on the increased difficulty of sludge dewatering after mixed treatment of leachate and municipal sewage, as well as the possible negative impact of the sludge initial pH. Finally, the effect of sludge initial pH on sludge bio-leachate dewatering in the range of 3.0~10.0 was studied by utilizing the feature that bio-leachate can be deeply dehydrated. The following results were obtained:

(1)    After bioleaching, the pH of the sludge decreased from (6.62~7.04) to (2.02~2.90), and the decrease rate was 56.06~70.07%, with the energy source reductive sulfur oxidation rate increasing by 34.64~63.28% and the $Fe^{3+}$ concentration increasing by 20.91~65.89%.

(2)    After bioleaching, the SRF of sludge decreased to the lowest value (0.12~1.43) $\times 10^{12}$ m/kg from (4.05~4.83) $\times 10^{12}$ m/kg. When the sludge initial pH was 5.0 and the leaching duration was 48 h, the SRF decreased from 14.50 $\times 10^{11}$ m/kg to 0.12 $\times 10^{12}$ m/kg, showing a 91.66% reduction, and the dewatering performance of the sludge reached an optimum state.

(3)    The dominant microorganisms did not show significant variation in relative abundance at the phylum level with variation in the sludge initial pH; however, the community did differ significantly at the genus level. When the initial pH value was 5.0, after 48 h of bioleaching, autotrophic bacteria that can oxidize $S^0$ and $Fe^{2+}$, namely *Thiomonas*, *Halothiobacillus* and *Acidithiobacillus*, dominated the community. The relative abundances of these genera were 4.91%, 2.79% and 0.80% respectively. At this time, the ORP and pH reached the maximum and minimum values, respectively, compared with the blank test results. This shows that the acidification and oxidation that occurs during the bioleaching of sludge may be due to the synergistic effects of such microbial community members, which was also the fundamental reason for the improved sludge dewatering performance.

**Author Contributions:** Investigation, J.W. and G.W.; software, G.W. and S.L.; writing—original draft, S.L.; writing—review and editing, M.S. and H.Z.; S.L. and M.S. equally contributed to the work. All authors have read and agreed to the published version of the manuscript.

**Funding:** This work was supported by the National Natural Science Foundation of China (51308136); the China Scholarship Council Young Core Instructor Overseas Research and Training Project (CSC201709945022); the Graduate Student Education Innovation Project of Guangdong Province; the Guangdong University Student Entrepreneurship Training Program; and the Guangzhou University College Student Innovation Training Program.

**Conflicts of Interest:** The authors declare no conflict of interest.

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
