# Peer review of "Influence of Sludge Initial pH on Bioleaching of Excess Sludge to Improve Dewatering Performance"

_coatings, doi:10.3390/coatings10100989_

Round 1
Reviewer 1 Report
Page 2, lines 53-54: Two exactly the same sentences, but 2 different references [15-17] and [18].
Paragraph 2.1., page 2: Please indicate the number of samples? Include the mean value +/- RDS in table 1.
Paragraph 2.2.1., page 2: Please include a reference to a standard method, or indicate if a laboratory-developed method was used.
Please write the full names of chemicals when the first appear in the text (e.g. page 3, line 120).
Page 4, table 2: Please refer to standard methods and give more information on the method and/or the apparatus used. Please write the full names of chemicals when the first appear in the text (e.g. VSS, TOC, etc.)
Page 13 onward: Please improve the quality of fig. 9-14 and 17-18 if possible.
Figure 2, page 6 and figure 4, page 8: For better visualization, please give a little bit of breading space between the graphs, or divide them to a), b) and c) for the different parameters.
Paragraph 6, page 20 line 358: Please present the actual values, rather than the percentage of reduction for the pH, it will give a better explanation of the effect. If you still want to show the percentage, include it into brackets.
References: Please edit the journal’s names according to the instructions for authors for the Coatings. Journals’ full names and journals’ abbreviations can not be used together.
Reviewer 2 Report
Dear Authors,
Your manuscript No. 931505 needs improvement. I have a few comments listed below. I hope these can help to improve your paper. I did not evaluate the part on genetic testing because it is beyond my research scope – I’m wastewater technologist not the biologist.
1) In my opinion the topic of your manuscript isn’t in the scope of the Coatings journal. Additionally in your manuscript there is no references from Coatings, perhaps because of the lack of papers in this topic. In Coatings I found 0 results for “excess sludge” keywords, 0 for “sludge”, 0 for “bioleaching” 1 for “sewage”, 3 for “wastewater” – but rather about processes on surfaces and films. So, I recommend changing the journal to e. g. Water (section Wastewater Treatment and Reuse).
2) The introduction is short but clear. One important mistake: the sentence “However …” from lines 52-54 is duplicated in lines 54-55. Unfortunately the sentences have different references! Are the references 15-17 properly used? They are about bioleaching, not about leachate management strategy in China.
3) The methodology needs to be reorganized. The description of main idea isn’t at the beginning. Perhaps the order 2.1, 2.4, 2.3, 2.2 will be better. Please consider that. Firstly you inform about the origin of the research sample – it’s ok. Next should be presented the main experiment (designation of dependent and independent variables, experimental conditions and information about samples type, way of taken and measured parameters). Next more detailed information about inoculum preparation and used research methods of DNA analysis and methods of wastewater characterisation.
4) The captions of equations 1-3 are shifted. Should the lines 114 and 115 be below eq.2 and lines 116 and 117 below eq. 3?
5) In the Table 2 you presented the methods, but you didn’t describe the measured sample and its preparation. E.g. the method with phenanthroline for Fe determination needs clear sample without turbid. The same SO42-. How did you prepared the sludge sample? How was with TOC analysis?
6) Table 2, bottom row – instead “natural settlement method” I propose “Gravity separation method”
7) The “initial pH” assumption is questionable. As I can see in the Figure 1, the initial pH for series “pH 10.0" was about 6, for “pH 8.0” about 5.5, etc. You alkalized the sludge or acidified and then mixed with acidic inoculum, so the system you investigated was characterized by intermediate pH of these two strains (sludge and inoculum). The pH values after mixing was the initial for the process. Therefore I think using the term “initial pH” in relation to pH of sludge is failed. The initial pH of the system should be considered after mixing. Two solutions with a given pH may have different resultant pH when mixed, it depends on solution composition, e.g. salinity. The results of your research can’t be generalized, because when I prepare inoculum like you and alkalize my sludge (from other A2/O plant – environmentally sample) to pH 10.0 and then mix, I probably obtain different pH of the mixture at the beginning than you (~6). Thus:
lines 146, 147: is present “The pH was controlled in the range of 3.0 to 10.0” – wasn’t
lines 158, 159: is present “The increase under acidic conditions was about 3.48%~11.09% higher than under alkaline conditions” – during bioleaching wasn’t alkaline conditions
the same line 172: “alkaline conditions”, etc.
8) At the final stage of bioleaching the SO42- concentration decreased (lines 173, 174 and Fig. 2). The same Fe3+ obtained during biooxidation. Why? Was it connected with ORP decrease at the final stage? The decrease of ORP was indicated in all series, so please explain it. It is important to optimization the time of bioleaching. What about oxygen? Reactions 4 and 5 indicate O2 consumption. What was the O2 concentration in the sludge? Whether it was sufficient to cover the O2 demand of biooxidation during 72 h?
9) If you use the abbreviation, please give the full term by first using.
10) line 71: is present A2 O-1 – it isn’t the unit but abbreviation and I recommend A2/O
11) To enrich the inference, I suggest making an analysis of variance and correlation of variables. Statistical analysis can improve yours research (see e.g. DOI: 10.1061/(ASCE)EE.1943-7870.0001008). In present form the dependencies are not confirmed (only intuitive) as well as it was often a discussion of how many% more, how many% less
12) Please unify the pH values: line 282 – “pH values of 3, 5 and 6.9”; line 285 and everywhere else – “3.0, 5.0 and 6.9”; Figure 12 – pH=3, pH=5, pH=7
13) In table 1 is “settlement rate” whereas in text is present “sludge sedimentation rate”.
14) Please add the explanation, why in experiments some blind series are omitted. It is especially important in tests showed in the figures 6 and 8.
For example:
the SRF of sample of sludge was (4.05-4.83)x1012 (table 1)
the SRF of sludge +inoculum was 4.1 x 1012 (figure 6, point at t=0 for series pH 6.9)
the SRF of sludge +inoculum + NaOH was 2.5-2.9 x 1012 (figure 6, point at t=0 for series pH 8.0 and 10.0)
the SRF of sludge +inoculum + HCl was 0.4-0.6 x 1012 (figure 6, point at t=0 for series pH 5.0 and 3.0)
Thus the addition of NaOH or HCl results decreasing of SRF at time 0h without bioleaching process. Ok, it is understandable and is related to the surface action, but the results from series “sludge + inoculum + NaOH/HCl" should be compared with blank series "sludge+NaOH/HCl". Then the experiment will be clear.
The same with the sludge sedimentation rate when analysing the data from table 1 and fig.8.
15) The procedure of EPS determination is more complex than “heat extraction”, especially when you analysed the EPS fractions and write “replaced by TOC/VSS ratio”. Additionally the paper you cited [22] isn’t in free access. The explanation is needed.
16) line 316: is “The number of heterotrophic bacteria shown in Fig. 8 decreased”, but I think in fig. 5
Additionally, I propose to give the results together with the results in fig. 5, e.g. in complex figure and discussion of them. The article is very long and it is hard to read when the data are so scattered.
Why in the methodology you omitted the information about blank tests while even present it in conclusion?
Best regards,
Reviewer
Reviewer 3 Report
“Influence of initial pH on bioleaching of excess sludge to improve dewatering performance” presents interesting findings on this topic. In my opinion, it can be published after major revisions have been implemented. My comments are the following:
Highlights should be rewritten. They should focus on the main findings of the research.
A recent paper on advanced treatments for sewage sludge minimization could be useful for you (https://doi.org/10.3390/app9132650)
Lines 52-52: “However, due to the limited disposal capacity of some landfills in China, it is necessary to mix landfill leachate with municipal sewage”. Leachate is unrelated to the limited or no disposal capacity of a landfill. Leachate is produced by the landfill whether it is active or not (even if with different characteristics). Please, amend this sentence.
Line 52: please, eliminate the second “.”
Line 54-55: repetition of line 52-53.
Line 60: Correct. The release of EPS could be a cause of foaming that could create big problems in wastewater and sewage treatment plants. This aspect should be highlighted in a sentence. Here two recent works that you could cite: https://doi.org/10.1007/s11356-020-09143-y https://doi.org/10.3390/app10082716
At the end of the Introduction, the novelty of this paper should be better highlighted.
Line 75: This table is referred to sludge after mix with leachate or not? Please, specify better this aspect.
Line 126: The size of “2.4.” is higher than other section headings.
Line 134: “(Replace by TOC and VSS ratio)”. Please, clarify this aspect.
In figures, please, use also different colours to better highlight differences.
Line 259: “Microbial community structure” results should be inserted on Section “Results and discussion”. Section 3, 4 and 5 should be merged (in alternative change the name of section 3).
In my opinion, several figures could be merged in only one figure and more subfigures (e.g. 10 and 11, 13 and 14, 16 and 17,…). Now the number of figures is very high.
In the conclusions the importance, the novelty and the applicability of your findings should be better highlighted.
Round 2
Reviewer 1 Report
Dear authors,
the revised manuscript addresses all the comments made and was improved. I find it suitable for publication in the present form.
Author Response
Dear Editor and reviewers:
Thank you very much for your review comments and support.
Reviewer 2 Report
Dear Authors,
You really improved your manuscript. Thank you for your work. I give you “minor revision” because finally I have got some important suggestion.
With regard to my comment No. 1: You did not explain why your manuscript addresses the subject of the Coatings journal. I still see your paper more in Water journal.
With regard to my comment No. 7 and your answer on it: In my opinion your approach is simplified, because from the technological point of view more important is pH during the process after mixing the strains (at beginning, middle and end) than pH of strains before mixing. However, if you don’t want to change your approach please change the nomenclature. I strongly suggest to replace the term “initial pH” with “sludge initial pH” in all text and captions. Then there will be no doubt.
I suggest to include your answers to my comment No. 8 and 15 in the manuscript text.
If you put to the manuscript the results of correlation analysis, then you shall to put the relevant information to the methodology. Needs to be completed.
You should more strongly emphasize in the text your response to my comment No.14. Also, your conclusions should be more general regarding acidification or alkalization of the sludge before bio-leaching.
Previously I omitted one question. You write about landfill leachate in Introduction, in Methodology and at the end even in Conclusions. It is the main part of your aim (lines 60-63), but unfortunately you don’t discuss the results in relation to the topic. If it is so important and you highlight this point in the objective and in results, you should more emphasize the importance of landfill leachate occurrence on the bioleaching process. Also the landfill leachate chemical characteristics (can be different depending on the type and age) is absolutely required.
Best regards,
Reviewer
Reviewer 3 Report
I see that the Authors implemented all my suggestions. Therefore, I think that this manuscript can be published.
Author Response

(The authors gave the same response as above.)
